# Haplotypes of ATP-Binding Cassette *CaABCC6* in Chickpea from Kazakhstan Are Associated with Salinity Tolerance and Leaf Necrosis via Oxidative Stress

**DOI:** 10.3390/biom14070823

**Published:** 2024-07-10

**Authors:** Gulmira Khassanova, Satyvaldy Jatayev, Ademi Gabdola, Marzhan Kuzbakova, Aray Zailasheva, Gulnar Kylyshbayeva, Carly Schramm, Kathryn Schleyer, Lauren Philp-Dutton, Crystal Sweetman, Peter Anderson, Colin L. D. Jenkins, Kathleen L. Soole, Yuri Shavrukov

**Affiliations:** 1Faculty of Agronomy, S.Seifullin Kazakh Agrotechnical Research University, Astana 010000, Kazakhstan; satidjo@gmail.com (S.J.); gabdolaadema@gmail.com (A.G.); happy.end777@mail.ru (M.K.); azailashova@gmail.com (A.Z.); 2A.I. Barayev Research and Production Centre of Grain Farming, Shortandy 021601, Kazakhstan; 3Faculty of Natural Sciences, Central Asian Innovation University, Shymkent 160000, Kazakhstan; kuntun-gulnar@mail.ru; 4College of Science and Engineering, Biological Sciences, Flinders University, Adelaide, SA 5042, Australia; carly.schramm@flinders.edu.au (C.S.); katie.schleyer@flinders.edu.au (K.S.); lauren.philpdutton@flinders.edu.au (L.P.-D.); crystal.sweetman@flinders.edu.au (C.S.); peter.anderson@flinders.edu.au (P.A.); colin.jenkins@flinders.edu.au (C.L.D.J.); kathleen.soole@flinders.edu.au (K.L.S.)

**Keywords:** chickpea, DArT analysis, gene expression, glutathione, haplotype, malondialdehyde, marker-trait association, salinity, SNP, oxidative stress

## Abstract

Salinity tolerance was studied in chickpea accessions from a germplasm collection and in cultivars from Kazakhstan. After NaCl treatment, significant differences were found between genotypes, which could be arranged into three groups. Those that performed poorest were found in group 1, comprising five ICC accessions with the lowest chlorophyll content, the highest leaf necrosis (LN), Na^+^ accumulation, malondialdehyde (MDA) content, and a low glutathione ratio GSH/GSSG. Two cultivars, Privo-1 and Tassay, representing group 2, were moderate in these traits, while the best performance was for group 3, containing two other cultivars, Krasnokutsky-123 and Looch, which were found to have mostly green plants and an exact opposite pattern of traits. Marker–trait association (MTA) between 6K DArT markers and four traits (LN, Na^+^, MDA, and GSH/GSSG) revealed the presence of four possible candidate genes in the chickpea genome that may be associated with the three groups. One gene, ATP-binding cassette, *CaABCC6*, was selected, and three haplotypes, A, D1, and D2, were identified in plants from the three groups. Two of the most salt-tolerant cultivars from group 3 were found to have haplotype D2 with a novel identified SNP. RT-qPCR analysis confirmed that this gene was strongly expressed after NaCl treatment in the parental- and breeding-line plants of haplotype D2. Mass spectrometry of seed proteins showed a higher accumulation of glutathione reductase and S-transferase, but not peroxidase, in the D2 haplotype. In conclusion, the *CaABCC6* gene was hypothesized to be associated with a better response to oxidative stress via glutathione metabolism, while other candidate genes are likely involved in the control of chlorophyll content and Na^+^ accumulation.

## 1. Introduction

Chickpea (*Cicer arietinum* L.) is an important crop in many countries. Kazakhstan is a moderate producer, with about 9.5 K tons of dry chickpea seeds harvested from 11.5 K hectares in 2022 [1]. There is, however, potential for expansion in chickpea cultivation in Kazakhstan as it is an increasingly valuable crop for human foods and fodder, with a rich nutritional profile of proteins, vitamins, dietary fiber, and many essential trace elements.

The production of chickpea as a legume crop is limited, however, by major threats like drought and salinity. This has become especially apparent due to the influence of climate change [2]. Salinity has only a localized and rather ‘patchy’ distribution in Kazakhstan, but tolerance to NaCl remains a very important factor for crop agriculture, including chickpea production. The production of salt-tolerant cultivars only is estimated as the most promising solution to growing crops in areas with high salinity [3].

Plant symptoms resulting from a high NaCl level are quite diverse, but well described and presented in practically every study of salinity stress in plants. However, chlorophyll degradation and Na^+^ accumulation could be among the simplest biochemical analyses to estimate plant salinity tolerance and sensitivity [4,5]. For example, chlorophyll content in salt-sensitive chickpea cv. Rupali was half that of tolerant cv. Genesis836, indicating significant chlorophyll degradation [6,7].

Additionally, leaf necrosis is one of the phenotypic traits easily evaluated visually and is scored as a proportion of dead and yellow leaves and stems to those still green and alive. The leaf necrosis score has been extensively used for the evaluation of salinity tolerance in various plant species, for example water hyacinth (*Eichornia crassipes*) [8], mulberry (*Morus* sp.) [9], and ornamental beardtongue (*Penstemon barbatus* and *P. strictus*), in urban landscapes [10]. In legume crops, an effective method for leaf necrosis score was developed and successfully used in soybean [*Glycine max* (L.) Merr.] [11] and faba bean (*Vicia faba* L.) [12]. In chickpea, 557 diverse accessions were screened in hydroponics with a sand- and gravel-based media, and leaf necrosis scores assessed for 21-day-old plants exposed to salinity levels of about 60 mM NaCl; the authors reported 33 accessions as tolerant to salinity [13]. In contrast, in chickpea plants tested in field trials, a leaf necrosis score was reported as a proxy for salinity tolerance, and a moderate negative correlation was found between leaf necrosis and seed yield (*R*^2^ = 0.15–0.31) for the cultivars Rupali and Genesis836 [14].

Oxidative stress is a consequential threat, and its elicitation always accompanies such stresses as salinity, where the increased level of reactive oxygen species (ROS) in plant cells is an unavoidable effect [15]. Therefore, plants are forced to manage ROS detoxification more effectively and faster to return to non-dangerous levels. There are many compounds that can be used as indicators for oxidative stress. One of the most widely used biochemical indicators of oxidative stress is malondialdehyde (MDA), which occurs as a product of lipid peroxidation in plant cells [16,17].

The biochemical mechanisms of oxidative stress are complicated but, in general, well described [18,19]. The glutathione pathway can play an important role in ROS detoxification in all living organisms, including plants [20,21]. The balanced cycles between glutathione in reduced form (GSH) and glutathione disulfide in oxidized form (GSSG) can indicate good oxidative stress management, keeping ROS under control [22]. The total content of glutathione in any plant organ, including leaves and seeds, also represents the potential for dealing with oxidative stress [23].

Enzymes involved in the glutathione pathway can be critical for the effective removal of harmful ROS in stress-affected plants, and the role of a few of these enzymes is emphasized here. Glutathione peroxidase (GPX) converts the ROS hydrogen peroxide to water and oxygen, by oxidizing GSH to GSSG [24]. Glutathione reductase (GR) recycles GSH from GSSG, but must oxidize NADPH to do so [17,25,26]. A high ratio of GSH/GSSG can indicate plants with better management of oxidative stress. Another enzyme, glutathione S-transferase (GST), catalyzes the conjugation of glutathione to a range of other chemical groups, with an important role in sequestration of toxic compounds into seed cell vacuoles [27]. Such glutathione conjugates include heavy metals like cadmium and mercury, chlorophyll catabolites, anthocyanins, peroxidized lipids, and some other compounds. Altogether, enzymes of the glutathione pathway work cooperatively in response to oxidative stress [22].

Hundreds to thousands of genes can be involved in the plant response to salinity and accompanying oxidative stresses [28,29,30,31,32], and many non-related accessions, landraces, breeding lines, and other genotypes of various species, including chickpea, have to be studied in different environments for phenotyping [33,34]. The results can be incorporated into existing or developing fine genetic maps for quantitative trait loci (QTL) identification [35,36,37]. In contrast, quantitative trait nucleotide (QTN) analysis reflects the association between a single nucleotide substitution or polymorphism (SNP) and the corresponding haplotype (a group of clustered SNPs in a certain genetic region of chromosome) for the studied traits [38,39]. This is also called MTA, marker–trait association, where SNP-based or other molecular markers can be studied using manual or low- or high-throughput genotyping methods [40,41,42,43]. Diversity array technology (DArT) is also used for QTL, QTN, and MTA in diverse plant species, including chickpea [44,45,46]. The identified MTA is linked with molecular markers in the corresponding genetic region containing a possible candidate gene. This is an important step for the more precise identification of candidate genes used in many reports, including chickpea [47,48,49].

Different options can be used to verify a candidate gene identified in the plant response to stresses. In the present study with chickpea, a candidate gene was analyzed for gene expression using RT-qPCR after NaCl treatment, which is now a ‘first-choice’ and routine method [50,51,52]. To provide stronger validation of the identified candidate gene, it was important to study the gene expression in chickpea parents and advanced hybrid breeding lines, as conducted previously, for example, in tomato (*Solanum lycopersicum* L.) [53] and maize (*Zea mays* L.) [54]. Proteomics is a method that can be used for the analysis of protein diversity in plants grown in various conditions. Comprehensive mass spectroscopy is a very powerful protein analysis technique, based on developed spectral libraries for fractions or unfractionated samples from different plant species, including chickpea [55,56]. In our experiments, mass spectrometry was applied to identify differential protein profiles in the mature seeds of chickpea accessions used in this study.

Here, we hypothesize that a transporter gene, ATP-binding cassette (ABC), associated with some identified SNP and haplotypes, could be one of the candidate genes involved in the response of chickpea plants to salinity and oxidative stress. This gene belongs to the very large and diverse family of ABC transporters, widely studied in various plant species [57,58,59,60]. The *CaABCC* family of genes in chickpea [61,62] have been characterized and shown to function in transporting glutathione from the cytosol to vacuoles, conjugated with cadmium, mercury, or other heavy metals [63,64,65,66]. Other functions may include herbicide detoxification [67,68], the transport of acylated anthocyanins [69,70], organic anion transport and chlorophyll degradation [71], folates and antifolates accumulation [72], stomatal opening during drought [73,74], and root development [75,76].

A candidate gene *CaABCC6* (Ca09705) was reported earlier, with SNPs and haplotypes in different chickpea accessions; it was found to be involved in the transport of glutathione conjugates and results in a different seed size and 100-seed weight [77]. In the present study, SNPs were also observed in this gene among chickpea accessions. Therefore, this was deemed an interesting gene to pursue in our study on plant responses to salinity and oxidative stress. Our hypothesis was tested by evaluating the expression of this gene in parents of hybrids and advanced hybrid breeding lines.

The aim of this research was to identify genes that relate to biochemical and physiological responses to salinity in diverse chickpea accessions. This was achieved by quantifying measures of salinity stress, including leaf necrosis, chlorophyll content (degradation), and Na^+^ accumulation after NaCl application; as well as two oxidative stress indicators, lipid peroxidation products and glutathione redox state in leaves. Marker–trait association was then conducted using a 6K DArT assay, ultimately leading to the determination of a key SNP and a novel haplotype identification in the *CaABCC6* candidate gene. Expression analysis of *CaABCC6* was then used to determine the relationship between gene expression and salinity tolerance, followed by a proteomics approach that demonstrates differences in glutathione-related protein levels in the mature seeds.

## 2. Materials and Methods

### 2.1. Plant Material

In this study, five chickpea accessions, Desi-type, ICC-1052 (Pakistan), ICC-5613 (India), ICC-10945 (India), ICC-11121 (India), and ICC-12654 (Ethiopia), were obtained from the ICRISAT Reference set collection (India). Seeds of four chickpea cultivars, Kabuli-type, were provided by S.Seifullin Kazakh AgroTechnical Research University, Astana, Kazakhstan. The first two cultivars originated from Russia, and their seeds are available in the Genebank germplasm collection of Vavilov Research Institute of Plant Genetic Resources [78], with presented catalogue numbers as follows: (1) Krasnokutsky-123 from Krasnokutsky Breeding Station, 1982 (K-1917, with dark seeds); and (2) Privo-1 from Volgograd Department of Research Institute of Plant Breeding and Seed Production, 1995 (K-3484, this and other cultivars with light seeds). The last two cultivars originated from Kazakhstan: (3) Tassay from Krasnovodopadskaya Breeding Station, 2008 (K-1965); and (4) Looch was from the Kazakh Research Institute of Agriculture and Plant Growing, 2008, and the Registration Patent is present on the web-site [79]. The choice of the studied chickpea accessions was based on the very diverse reactions of plants to stressful conditions in the preliminary experiments in the hybridization program.

Two hybrid populations were established with details presented in a previous paper [80]: (1) ♀Krasnokutsky-123 × ♂ICC-12654, and (2) ♀Tassay × ♂ICC-1052. Originally, 12 F_6_ breeding lines were developed in each population, coded H18 and H23, respectively, and three breeding lines were selected from each population for qPCR analysis.

Seeds were sown in 18 cm diameter pots with 2.6 kg of soil-mix (BioGro, Adelaide, Australia), four seedlings per pot, with artificial inoculation of rhizobium NodulN, chickpea group N (New Edge Microbials, North Albury, NSW, Australia). Plants were grown in pots with soil for one month in a controlled-temperature greenhouse with 25 °C /20 °C day/night temperature and 16 h LED Grow Lights (~PAR 500) (Heliospectra AB, Gothenburg, Sweden). Pots were watered twice weekly on a portable scale, keeping soil moisture level consistent at 80% field capacity.

### 2.2. Salinity Stress Indicators: Leaf Necrosis and Chlorophyll Content

Salt stress was applied to one-month-old plants similar as described earlier [81], and 150 mL of 200 mM NaCl was added to each pot, with four increments, twice daily over two days. The calculated level of salinity in the experiment reached 100 mM NaCl and this was maintained until the end of the experiment. In control pots, the same volume of tap water without NaCl was used under the same schedule.

Leaf necrosis was scored visually after 21 days since the start of NaCl application using a scale of 1–10, where ‘1′ = a green and healthy plant with no salinity symptoms, ‘5′ = leaves on the bottom half of the plant are showing necrosis but leaves on the top half of the plant are yellowing or green, and ‘10′ = leaves and stems are completely necrotic and dead [13].

All other experiments were carried out 15 days after the start of NaCl treatment. Chlorophyll measurements were conducted on the young fully developed expanded leaf, about 5 cm from the tip of the shoot, using a SPAD meter (Soil plant analysis development, Model 502plus, Konica Minolta, Singapore) and expressed as SPAD value units following the description published earlier [7]. Chlorophyll content was checked in plants of all studied chickpea before the experiment to eliminate potential variations in leaf structure, thickness, and chlorophyll distribution.

### 2.3. Na^+^ and K^+^ Accumulation and Ratio Na^+^/K^+^ in Leaves

All leaves from two main shoots, 5 cm from the top, were pooled from each plant, making one sample. Three samples were collected from three plants of each genotype representing three biological replicates. FW was recorded and samples were dried for two days at 80 °C. Leaf tissue sap (FW-DW) was used for determination of Na^+^ and K^+^ levels and for their ratio of Na^+^/K^+^. Leaf samples were digested in 10 mL of 1% HNO_3_ at 80 °C for 4 h. Concentrations of sodium and potassium ions were measured by Flame-photometer (Sherwood, UK, model 420), compared with calibrating standards with known content of NaCl and KCl, and expressed as concentration (mM in plant sap), following the previously published description [81].

### 2.4. Oxidative Stress Indicator: MDA Content

Lipid peroxidation was measured as malondialdehyde (MDA) equivalents using the thiobarbituric acid reactive substances (TBARS) assay following a previously published protocol [82] with some modifications. Frozen leaf tissue was homogenized in liquid nitrogen, and 0.1 g was transferred into a 1.5 mL microtube, mixed with 1.0 mL 80% ethanol and centrifuged at 13,500× *g* for 15 min. Two aliquots of supernatants, 350 µL each, were transferred to new separate microtubes and equal volume of either Solution 1 or 2 were added and mixed. Solution 1 contained 20% trichloroacetic acid (TCA), 0.01% butylated hydroxytoluene (BHT) in ethanol, and 0.5% thiobarbituric acid (TBA). Solution 2 was exactly the same but with exclusion of TBA. All samples were then heated at 96 °C for 30 min, rapidly cooled on ice for 5 min, and centrifuged at 13,500× *g* for 15 min. Absorbance of the supernatants (100 μL) was measured at: 532 nm (TBA-MDA complex); 600 nm (nonspecific turbidity); and 440 nm (interfering soluble sugars), using a microplate reader (CLARIOstar Plus, BMG Labtech, Ortenberg, Germany). MDA equivalents were calculated as previously described [82], where MDA component A = [(Abs532 + TBA − Abs600 + TBA) − (Abs532-TBA − Abs600-TBA)]; MDA component B = [(Abs440 +TBA − Abs600 + TBA) × 0.0571]; and final MDA equivalents (nmol/mL) = [(A − B)/157,000] × 10^6^. In this method, MDA content was measured with and without TBA to increase the accuracy of MDA measurements by correcting for any non-MDA compounds that also absorb at 532 nm, such as anthocyanins.

### 2.5. Content of Glutathione (GSH) in Reduced Form and Glutathione Disulfide (GSSG) in Oxidized Form and Their Ratio (GSH/GSSG) in Leaves

The contents of glutathione GSH and GSSG and their ratio were measured as described previously [83] with recent modification [84] and adjusted for the current experiment. For each sample, two leaves from one plant were frozen and ground in liquid nitrogen, via vortexing with 8 mm ball-bearings. A portion of frozen leaf powder (0.2 g) was extracted in 1 mL of cold 5% trichloroacetic acid (TCA) at 4 °C on a rotary shaker for 15 min with subsequent centrifugation at 12,000× *g* for 20 min. The supernatant was used for the enzymatic assay. Firstly, for the total glutathione assay, 100 µL of extract was mixed with 100 μL of sterile H_2_O. For the GSSG assay, another 100 µL of extract was mixed with 2-vinylpyridine to mask GSH and incubated at 25 °C for 1 h. The reaction mixtures (1 mL) contained 0.1 M phosphate buffer pH 7.5, 1 mM 5,5’-dithiobis- 2-nitrobenzoic acid (DTNB), 1 mM NADPH, 1 U of glutathione reductase (Sigma-Merck, St. Louis, MO, USA), and 100 μL of the tissue extract either for GSH or for GSSG. The concentrations of GSH and oxidized glutathione GSSG were determined via absorbance at 405 nm using a spectrophotometer (UV-3100 PC, VWR International, Leuven, Belgium). GSH content was calculated from the difference between the concentrations of total and oxidized glutathione, with following calculation of the GSH/GSSG ratio in each sample.

### 2.6. 6K DArT Assay

DNA was extracted from bulked leaf samples from five plants in each accession grown in non-stressed (control) conditions, using a standard phenol–chloroform method [85] with minor modifications [80]. After, DNA (50 µL aliquots of 100 ng/µL) was submitted to Diversity Array Technology Co., Canberra, Australia [86] for genotyping using chickpea DArTseq (1.0) with 6K DArT clones. Fluorescently labelled clones representing the entire chickpea genome were hybridized, printed, and scanned on a microchip. The DArT markers score is based on presence or absence of the signal and designated as ‘1′ or ‘0′, respectively. Results of Silico-DArT analysis with mapping of all identified SNP were retrieved and used for further study. All identified DArT markers went through several steps of filtration to remove redundant SNPs and those with unknown mapping in the chickpea reference genome.

### 2.7. Marker–Trait Association (MTA)

The marker–trait association analysis between all DArT markers passed through filtration for quality control, then four morphological and biochemical traits (leaf necrosis, Na^+^ accumulation, MDA, and GSH/GSSG ratio) were tested using the mixed linear model (MLM) with kinship matrix and Q matrix in TASSEL 5.0 [87]. MLM (K+Q) can reduce type I errors compared to the General Linear Model (GLM) and prevent more false positives. Markers with minimum frequency of alleles less than 5% were excluded and the kinship matrix was obtained based on all DArT markers using TASSEL 5.0 as described earlier [46]. Legume Information System (LIS) was used for confirming the identification of candidate genes and their location in the annotated genome of chickpea cv. Frontier, v1.0 assembly [88].

### 2.8. DNA Extraction, Sanger Sequencing, and SNP Identification

DNA was extracted as described in Section 2.6. Primers were developed to amplify fragments within the open reading frame (ORF) of candidate genes (Appendix A). PCR conditions and purification of the amplified PCR products were as previously reported [89]. The concentrations of purified PCR products were measured using NanoDrop spectrophotometer (Thermo Fisher Scientific, Waltham, MA, USA) and used as a template for Sanger sequencing at the Australian Genome Research Facility (AGRF), Adelaide (Australia). SNP were visualized using the Chromas computer software program, version 2.0 with manual comparison and SNP identification.

### 2.9. RNA Extraction and RT-qPCR Analysis of Gene Expression in Parental and F_6_ Breeding Lines

Chickpea plants were grown in pots with 100 mM NaCl application and control conditions over 15 days as described in Section 2.2. Two leaves, 5 cm from the top, were collected and pooled from each individual plant as leaf samples, then frozen and ground in liquid nitrogen. TRIzol-like reagent was used for RNA extraction following the protocol developed earlier [90]; then, cDNA synthesis and RT-qPCR analysis were performed as described previously [81]. Briefly, reverse transcription was carried out with 2 μg of RNA pre-treated with DNase (NEBiolab, Hitchin, UK) using the Protoscript Reverse Transcriptase kit (NEBiolab, Hitchin, UK) and cDNA samples were diluted with sterile water (1:10), ready for use in RT-qPCR analysis. KAPA SYBR Fast Universal Mix (KAPA Biosystems, Wilmington, MA, USA) was used in 10 µL total volume containing 0.5 µM primers and 3 µL of cDNA, and run in a CFX96 Real-Time qPCR system (BioRad, Hercules, CA, USA). Thermal cycling conditions were as described earlier [81]. Expression levels of target genes were normalized using the reference gene *CaELF1α*, elongation factor 1-alpha (AJ004960) [91]. Sequences of primers for targeted *CaABCC6* (Ca09705) and reference genes are present in Appendix A. At least three biological replicates (individual plants) and two technical repeats were used for each genotype and treatment.

### 2.10. Seed Protein Extraction, Purification, and Mass Spectrometry for Measurement of Enzymes from the Glutathione Pathway

Total soluble protein was extracted from mature chickpea seeds based on the original protocol [92] with the following modifications [93]. Briefly, for each sample, 0.3 g of crushed seeds was transferred to a 2-mL microtube with 2 mL 0.02% NaOH and incubated with shaking for 1 h at room temperature. Samples were centrifuged at 8000× *g* for 10 min and the supernatant was filtered through glass wool. The filtered extracts were freeze-dried overnight at −80 °C under vacuum and protein pellets were dissolved in 200 µL of 20 mM TRIS (pH 8.0) overnight at room temperature. Finally, extracts were centrifuged for 40 min at 40,000× *g* using an ultra-centrifuge (Optima Max-TL, Beckman, Indianapolis, IN, USA) and the clear supernatant was used for proteomics.

Proteomics analysis was carried out in the Omics Facility, Flinders University (Australia). The extracted protein samples were purified using a mixture of hydrophobic and hydrophilic SpeedBeads (Cytiva, Marlborough, MA, USA) then digested with trypsin, followed by clean-up and quantification as described earlier [94]. Mass spectrometry was carried out following the previously published protocol [95] with minor adjustments. In brief, the digested proteins were analyzed with a Dionex Ultimate 3000 uHPLC (ultra-high performance liquid chromatography) coupled to a Mass spectrometer (Fusion Lumos, Thermo Fisher Scientific, Waltham, MA, USA). An ‘in-house’ pulled column packed with 1.9 μm beads to 25 cm was used with 1.9 μm × 2 cm Nikkyos Technos Acclaim PepMap trap columns (Thermo Fischer Scientific, Waltham, MA, USA). Both solvents A and B, 0.1% formic acid in water or 80% acetonitrile, respectively, were used for peptide separation, washing, and equilibration.

To prepare a spectral library of all proteins, samples were pooled together and gas phase fractionation was applied targeting only a fraction of the expected mass range in each injection. Peptides from each sample were injected six times, to reduce complexity and increase depth of coverage. Data-independent acquisition was analyzed using Spectronaut software (https://biognosys.com/software/spectronaut, accessed on 30 June 2024) [96] for both spectral library generation and data analysis. The initial *p*-values were corrected to *q*-values as the final statistical differences. The level of *q*-value < 0.05 was designated as statistically significant, and the following levels were considered: * 0.01 < *q* < 0.05; and ** 0.001 < *q* < 0.01. The differences between expressed proteins were reported as both the percentage and Log2 changes in one group relative to another.

### 2.11. Statistical Analysis

Means, standard errors, and significance levels were calculated using unpaired *t*-test, ANOVA, and Pearson’s correlation functions based on software packages of Excel 365, Microsoft, and SPSS 25.0.0.0, IBM. At least three biological replicates (individual plants) and two technical repeats (instrumental runs) were used for each genotype and experiment.

## 3. Results

### 3.1. Salinity Stress: Leaf Necrosis, Chlorophyll Degradation, and Na^+^ Accumulation

Chickpea plants were treated with 100 mM NaCl and after 15 days of exposure to salinity, and symptoms of stress in the form of chlorophyll degradation were clearly observed. The evaluation of the results of the leaf necrosis (LN) scores and the measurement of chlorophyll content are present in Table 1. Three groups of genotypes with significant differences for both traits were consistent, while the poorest results with the highest LN score and the lowest chlorophyll content were recorded for five Desi ecotype ICC accessions (ICC-1052, ICC-5613, ICC-10945, ICC-11121, and ICC-12654). In contrast, two cultivars, Krasnokutsky-123 and Looch, showed a very low LN score and a significantly higher chlorophyll content. Two other cultivars, Privo-1 and Tassay, were in an intermediate position but still differed significantly from the two other groups (Table 1).

The dynamics of chlorophyll content and its degradation in the plants of six selected accessions with contrasting reactions to salinity (Figure 1A) showed very clear and significant differences. ICC-10945 and ICC-12654 performed poorly compared to the other accessions from as early as 6 days after NaCl application, while Privo-1 and Tassay began to show significant differences to the top performers from Day 12. Krasnokutsky-123 and Looch maintained chlorophyll content levels higher than the other accessions throughout the 15-day time course (Figure 1A). Control plants in all accessions had a very similar chlorophyll content during the experiment on the level of Day 0.

After six more days of NaCl treatment (21 days of salt stress, in total), the chickpea plants of accessions and cultivars with the most contrasting reactions to salinity were photographed. The plants of ICC-10945 and ICC-12654 looked almost completely dead, whereas the plants of Looch and Krasnokutsky-123 were affected but still had many green leaves and shoots (Figure 1B,C).

For Na^+^ accumulation (Figure 2A), all five ICC accessions had high and very high levels of sodium concentration in the sap of the leaf samples tested. The lowest level of Na^+^ in leaves was found in cultivars Krasnokutsky-123 and Looch, and varied between 32.9 and 35.6 mM, which is significantly different from all other accessions. The plants of two other cultivars, Privo-1 and Tassay, accumulated a moderate concentration of Na^+^ in leaves (Figure 2A). The ratio of Na^+^/K^+^ in the same samples showed a very similar pattern (Figure 2B), which may indicate that potassium concentration did not make a big difference to the contrasting response.

### 3.2. Oxidative Stress Indicators: Malondialdehyde (MDA) and Ratio between Reduced Glutathione (GSH) and Oxidized Glutathione (GSSG) in Leaves

Under salinity stress, the pattern of lipid peroxidation products (measured as MDA equivalents) in leaves showed significant differences between chickpea plants from the three groups described above in Section 3.1. Five ICC chickpea accessions showed a very high lipid peroxidation (Table 2), especially in contrast to the cultivars Krasnokutsky-123 and Looch. The cultivars Privo-1 and Tassay had intermediate levels of lipid peroxidation that were still significantly different from the other two groups (Table 2).

The calculated ratio between reduced and oxidized glutathione (GSH/GSSG) after 15 days of salt treatment showed the exact opposite trend, whereby the five ICC accessions had lowest GSH/GSSG ratio, and Krasnokutsky-123 and Looch had a significantly higher ratio, approximately two-fold compared to the lowest group. Again, Privo-1 and Tassay showed an intermediate phenotype (Table 2).

### 3.3. Marker–Trait Association Analysis for Salinity and Oxidative Stresses Using 6K DArT Assay

DArT-seq analysis with 6000 markers was applied to study genetic polymorphism and variability among a set of chickpea accessions. After an initial filtration, 3600 DArT markers were polymorphic, and 1600 of these had known mapping locations in the chickpea genome based on the publicly available physical map of cv. Frontier. The marker–trait association (MTA) analysis between 1600 filtered DArT markers and the morphological traits for salinity stress (leaf necrosis and Na^+^ accumulation) and for oxidative stress (content of MDA and GSH/GSSG ratio) in the studied chickpea accessions had similar patterns, and the example of MTA for leaf necrosis is presented in Figure 3 and more data in Table 3.

In the Manhattan plot, four significant and highly significant MTAs were identified and linked with DArT markers in chromosomes Ca2, Ca4, and Ca5, based on the physical map of the chickpea reference genome cv. Frontier, v.1.

Details of the identified DArT markers in each MTA and the closest most suitable candidate genes are summarized in Table 3, indicating differences between the studied chickpea genotypes. 

In four identified MTA, all studied traits showed high and very high scores indicating for a tight link between symptoms and traits of salinity (leaf necrosis and Na^+^ accumulation) and oxidative stress (malondialdehyde content and GSH/GSSG ratio). Therefore, all four potential candidate genes can be considered as possible important genes involved in the plant response to both salinity and oxidative stresses. The raw data for DArT microarray analysis are presented in Appendix A.

Only one Ca09705 gene (*CaABCC6*) was selected for further analyses based on published information that it was involved in glutathione transport in chickpea plants in non-stressed conditions [77].

### 3.4. SNP and Novel Haplotype Identification in the CaABCC6 Gene in Three Groups of Chickpea Genotypes Using Sanger Sequencing

In the results of Sanger sequencing analysis, seven SNP were found in the coding region of the *CaABCC6* gene in all studied chickpea accessions, including five Desi ecotype accessions (ICC-1052, ICC-5613, ICC-10945, ICC-11121, and ICC-12654), and four cultivars from Kazakhstan, Krasnokutsky-123, Looch, Privo-1, and Tassay. Examples of two important SNPs in two different accessions are presented in Figure 4.

A summary of the results for all SNPs in all the chickpea genotypes examined are presented in Figure 5. All five Desi ecotype accessions (ICC-1052, ICC-5613, ICC-10945, ICC-11121, and ICC-12654) showed identical sequencing of the studied fragments of *CaABCC6*, and their status for haplotype A was confirmed. In contrast, there were no haplotypes B or C identified among the studied genotypes.

Based on the sequencing results for SNP-3, -4, -5, and -7, indicated by green boxes in Figure 5, all four chickpea cultivars, Krasnokutsky-123, Looch, Privo-1, and Tassay, were initially identified as haplotype A. Importantly, SNP-3 and -4, are synonymous with no changes in amino acid sequence, whereas SNP-5 and -7 are non-synonymous and missense, encoding changes from haplotype A to B/C, corresponding to amino acid Phe968Ser and Phe984Val substitutions, respectively.

However, according to the sequencing results for SNP-6, all four cultivars have identical SNPs (indicated by red box in Figure 5) and, therefore, must belong to haplotype C. This SNP-6 is synonymous and does not change the encoded amino acid. Therefore, a new haplotype D is, for the first time, described here for the four studied chickpea cultivars, which can resolve this conflicting situation.

Additionally, our further study of the four cultivars revealed that haplotype D is still polymorphic. A novel SNP-1 was identified in the current study and is indicated by the blue box in Figure 5. This SNP-1 was unique in two cultivars, Krasnokutsky-123 and Looch, and differs from all others studied earlier and currently. In this regard, the new haplotype D was split into two, haplotypes D1 and D2, based on the unique SNP-1. This SNP-1 was a missense mutation and caused the change Glu699Lys only in haplotype D2, and in two cultivars, Krasnokutsky-123 and Looch, with this rare haplotype, differing from two other cultivars, Privo-1 and Tassay, with haplotype D1, as well as from other studied chickpea accessions.

### 3.5. CaABCC6 Gene Expression in Parents and F_6_ Breeding Lines

For gene expression analysis, parents of two hybrids, 1 and 2, as well as three F_6_ breeding lines from each hybrid, were selected based on their haplotypes of the *CaABCC6* gene. In these hybrids, the maternal parents, Krasnokutsky-123 and Tassay, had haplotypes D2 and D1, respectively, whereas both paternal accessions, ICC-12654 and ICC-1052, belonged to haplotype A (Figure 6). Two breeding lines from each of two hybrids, H18-2 and H18-3 (Hybrid 1), and H23-1 and H23-2 (Hybrid 2), inherited their maternal haplotypes of *CaABCC6*, D2 and D1, respectively. Breeding lines H18-1 and H23-3 from the same hybrids 1 and 2 had haplotype A, similar to those in their paternal parents, ICC-12654 and ICC-1052, respectively.

Clear discrimination was found in the expression analysis of *CaABCC6* in the plants of parental and selected F_6_ breeding lines in hybrids 1 and 2 under salinity stress (Figure 6). In hybrid 1, the differential expression level of *CaABCC6* in parental and breeding lines with haplotypes D2 started after five days of NaCl treatment and progressively increased after that. In contrast, plants with haplotype A showed an unchanged expression level of *CaABCC6* after five and seven days, but it increased after nine days of salt application. Nevertheless, even in the latter case (nine days), the *CaABCC6* expression was about three-fold higher in the plants of haplotype D2 compared to those of haplotype A. In hybrid 2, a very similar expression pattern of *CaABCC6* was observed but significant differences were recorded only after nine days of NaCl treatment between plants with haplotypes D1 and A, with about a two-fold difference in expression levels (Figure 6).

### 3.6. Mass Spectrometry Measuring of Glutathione Peroxidase (GPX), Glutathione Reductase (GR), and Glutathione S-Transferase (GST) in Mature Seeds

Protein compositions and their quantitative relationships were analyzed in mature seeds among nine initial chickpea accessions, using mass spectrometry. From the raw data of 7740 proteins, only three polypeptides were selected for the purposes of this study (Table 4). These polypeptides included three key enzymes for the glutathione pathway as follows: (1) Glutathione peroxidase, GPX (Q8L5Q6); (2) Glutathione reductase, GR (A0A1S3E1Y6); and (3) Glutathione S-transferase, GST (A0A1S2XSM5). The quantitative estimation of these three proteins is presented in Table 4 with the comparison to one chickpea cv. Looch with *CaABCC6* haplotype D2. There were no differences found between cultivars Looch and Krasnokutsky-123 (both with haplotypes D2) and, therefore, only data for cv. Looch are presented here.

For GPX content in matured seeds, all seven chickpea genotypes showed significantly higher level of GPX compared to cv. Looch with *CaABCC6* haplotype D2. However, a significant increase of 87.8–108.6% was recorded for two cultivars, Privo-1 and Tassay, whereas highly significant differences in the range of 179.5–288.2% were identified in five ICC accessions (Table 4).

A very different pattern of results was found for GR content, where it was significantly smaller in all seven chickpea accessions compared to cv. Looch. However, the differences were quite diverse among studied genotypes. For example, four out of five ICC accessions with *CaABCC6* haplotype A accumulated a GR much lower, in the range between −64% and −91.6%, with highly significant differences compared to those in cv. Looch with the *CaABCC6* haplotype D2. Smaller but still significant differences were found in two cultivars, Privo-1 and Tassay, with haplotype D1, and one accession (ICC-5613) with haplotype A varied between −41.9% and −48.2%, compared to cv. Looch (Table 4).

GST accumulation was similar to GR and significantly smaller than in cv. Looch, varying between −37.1% and −55.7%. Only one chickpea accession, ICC-11121, showed a highly significant reduction in GST, of −73.6%, compared to cv. Looch (Table 4).

## 4. Discussion

The response of plants to salinity is very complex and mitigating the processes of plant adaptation to short- and long-term NaCl stress is crucial. Oxidative stress represents a secondary challenging obstacle for plant growth, but it accompanies all types of abiotic and biotic stresses, where the production of ROS unavoidably occurs. Plants have to be tolerant to both salt and oxidative stresses, and their sensitivity to either type of stress can result in the suppression of their growth or even to plant death in extreme cases [97].

Many genes are involved in this plant reaction to salinity and oxidative stresses, but these genes belong to different pools. Salinity-responsive genes are involved in plant management of toxic concentrations of sodium and chloride as well as osmotic regulation, whereas genes controlling the detoxification of harmful ROS help with oxidative stress. Salt-tolerant plants can effectively regulate and coordinate gene activities from both pools [28,98].

In the present study, five chickpea accessions and four cultivars were studied that had contrasting reactions to salinity. The LN trait was strongly correlated with chlorophyll content, indicating damage in response to salt stress (Table 1 and Figure 1). This finding is in consensus with all previous similar studies of chickpea and other plant species [6,8]. LN can be easily observed visually, and it is used as a simple criterion for plant salinity tolerance [14], including in the present study. Three groups of chickpea genotypes were identified based on LN and chlorophyll content, and they were associated with Na^+^ accumulation in leaves and Na^+^/K^+^ ratio (Figure 2), which confirms our previous results [81] and does not differ from other published data [11].

The MDA content and glutathione ratio GSH/GSSG are popular and relatively simple indicators of oxidative stress. For example, a saline–alkali treatment using a 160 mM mixture of four salts (sodium chloride, sulfate, carbonate, and bicarbonate) in Chinese globeflower, *Trollius chinensis*, showed a consistent increase in MDA (2.3-fold) and total GSH content (3.7-fold) [99]. Similar results were reported for rice plants (*Oryza sativa* L.), cv. Tongxi926, grown in soil with high saline-alkalinity [100]. In both of these examples, plants were forced to manage with combined salinity, high pH, and oxidative stresses.

In the current study, the five chickpea accessions showed the poorest tolerance to salinity (high LN, low chlorophyll content, and high Na^+^ accumulation) and more sensitivity to oxidative stress (high MDA and low GSH/GSSG). In contrast, another group of two chickpea cultivars, Krasnokutsky-123 and Looch, were found to have much better tolerance to NaCl and less severe symptoms of oxidative stress. A third group containing two cultivars, Privo-1 and Tassay, showed intermediate symptoms of salinity and oxidative stresses. This kind of genetic polymorphism seen among chickpea accessions was reported earlier in different plant species with a tolerance to both salinity and oxidative stresses [27]. For example, diverse chickpea genotypes showed differential responses to NaCl application and subsequent oxidative stress according to their varied MDA content and glutathione pathway enzyme activities [17,25] as well as seed quality [101]. In soybean (*Glycine max* L.), two cultivars, JD19 and LH3, were found to grow better under salinity with a lesser MDA content compared to salt-sensitive cv. LD2 [102].

More intriguing results were obtained using MTA based on 1600 effective DArT markers. Four marker–trait associations and corresponding QTNs were found among the nine studied chickpea genotypes (Figure 3 and Table 3). It is very likely that all identified candidate genes are important, acting together for better plant response and adaptation to stressful conditions; therefore, we plan to analyze all of them in a future study. Some of the most suitable identified genes were directly stress-responsive, like heat-shock protein 80, *CaHSP80* (Ca17680). In *Arabidopsis thaliana*, the orthologous genes for *CaHSP80* were identified as three closely located *AtHSP81* genes (At5g56000, At5g56010, and At5g56030), and they were found and described as early responsive genes to dehydration and heat in RNA-binding proteome analysis [103] as well as during microarray differential expression between *A. thaliana* and the halophyte species salt cress (*Thellungiella halophila*) [104]. Another gene, *CaPPR*, Pentatricopeptide repeat (PPR)-containing protein (Ca04289) with ortholog in *A. thaliana*, At2g16880, was also described as an RNA-binding protein [103].

The third candidate gene was *CaPSII*, Photosystem II 22 kDa protein (Ca12664), putatively involved in the non-photochemical quenching mechanism of chlorophyll fluorescence based on the ortholog in *A. thaliana*, At1g44575. In *Arabidopsis*, this gene product is involved in support of the trans-thylakoid proton gradient and protection from photo-oxidative damage [105]. This ortholog gene, At1g44575, is also detected as an RNA-binding protein [103].

The fourth candidate gene was an ATP-binding cassette gene, denoted *CaABCC6* (Ca09705) (Table 3). There are many ATP-binding cassette genes in chickpea and other plants, some of which are also expressed after NaCl treatment, often concomitantly with other genes [106,107], including *HSP* genes, similar to those identified as genes of interest in the current study [108]. Some ABC genes participate in the glutathione pathway in plant cells by facilitating the transport of glutathione and glutathione conjugates [109,110,111]. It was therefore hypothesized that *CaABCC6* can also play an important role in the tolerance of chickpea plants to oxidative stress. To test this hypothesis, the gene *CaABCC6* (Ca09705) was selected and analyzed in the current study.

The genetic polymorphism of the gene *CaABCC6* was very well studied and published recently [77], where authors reported three haplotypes (A, B, and C) based on the sequence of the entire chickpea genome using the Illumina microarray platform. A more traditional approach based on Sanger sequencing was used in our current study, targeting ORF regions of *CaABCC6* in the collection of nine chickpea accessions and cultivars (Figure 4). All five ICC accessions confirmed their *CaABCC6* haplotype A, while the four chickpea cultivars belonged to another haplotype, D, as proposed in the present study. This likely happened because Basu et al. [77] studied the ICRISAT chickpea core collection and did not include modern chickpea cultivars from other countries. This point can be also illustrated by the published results based on another set of chickpea accessions grown in Indian locations, where another very different ABC transporter gene (Ca03567), orthologous to *AtABC1* (=*AtATH10*), was identified among seven candidates associated with the 100 seed weight trait [112]. Therefore, quite different genes could be identified depending on the plant genotypes of the studied accessions and their growing conditions.

A novel mutation in the *CaABCC6* gene was identified, providing SNP-1 in the position 2095 bp from the ‘Start’-codon. This SNP was found in two chickpea cultivars, Krasnokutsky-123 and Looch, resulting in Glu699Lys substitution in the encoded protein. This led to their designation under the novel haplotype D2 of the *CaABCC6* gene (Figure 5). Similar results were found with SNP identification using DArT study in African yam bean [*Sphenostylis stenocarpa* (Hochst ex. A. Rich.) Harms], where specific SNP alleles of two ABC transporter genes, homologs Phvul.006G053800 and Phvul.007G280200 from the *Phaseolus vulgaris* reference genome, were associated with seed size and weight [113].

In the present work, chickpea plants with the *CaABCC6* haplotype D2 (cultivars Krasnokutsky-123 and Looch) showed significantly better performance in all traits related to salinity tolerance, namely, lower LN, higher chlorophyll content, and lower Na^+^ accumulation and Na^+^/K^+^ ratio, as well as promising indicators for easier management with oxidative stress (a lower MDA and higher GSH/GSSG ratio), compared to those with haplotype D1 and especially compared to those with haplotype A. Additionally, a difference in seed setting and yield at harvest was observed among studied chickpea accessions, which was highest in the group with the *CaABCC6* haplotype D2 and lowest in those with haplotype A. However, this observation must be validated in further experiments with an appropriate number of biological replicates. It is also very important to extend this study and verify the identified haplotypes in more chickpea germplasms with diverse origins and plant reactions to salinity.

It is possible that any association between plants with *CaABCC6* haplotypes D2, D1, and A, and their response to salt and oxidative stresses is co-incidental. However, our results for the *CaABCC6* gene expression in parental and selected F_6_ breeding lines clearly indicated a significant correlation between haplotype and level of gene expression. Additionally, such significant differences for the *CaABCC6* gene expression were not only observed in parents but also inherited by F_6_ breeding lines with either the maternal or paternal type of the *CaABCC6* haplotypes and strongly correlated with levels of the gene expression under salt stress (Figure 6). A similar pattern of differentially expressed *ABCC* genes was reported in 12 accessions of a very popular Himalayan medicinal perennial plant species (*Picrorhiza kurroa* Royle ex Benth), where *PkABCC1* and *PkABCC2* genes were specifically expressed in stolon tissues and roots of the identified accessions [114].

Mass spectrometry of proteins extracted from mature seeds of all nine studied accessions was conducted. The results for three selected enzymes from the glutathione cycle pathway revealed very different but consistent tendencies among studied chickpea accessions (Table 4). The level of GPX proteins was very high in the seeds of chickpea cultivars and accessions with *CaABCC6* haplotypes D1 and A, compared to very low level of GPX in those with haplotype D2. In contrast, levels of GR and GST were significantly lower in seeds of the studied chickpea accessions with haplotypes D1 and A compared to the very high level of these enzymes in those with haplotype D2. That is, the cultivars with low levels of oxidative stress indicators showed lower levels of GPX and higher levels of GR, potentially explaining the increased GSH/GSSG ratio and level of GST protein. We must also mention that the presented proteomic profiles were studied in matured seeds but not in leaves or other plant organs after salinity-oxidative stresses. Nevertheless, the presented results for key enzymes from the glutathione cycle pathway in seeds (Table 4) might be interesting to illustrate our hypothesis for a better explanation of a potential role of the *CaABCC6* gene in glutathione conjugate in response to oxidative stress.

Two questions arise in the current study. Why is the ATP-binding cassette gene *CaABCC6* responsive to salinity? And how is this related to oxidative stress? Trying to address these questions, our working hypothesis is described below and visualized in the schematic diagram in Figure 7.

In most chickpea accessions with the *CaABCC6* haplotype A (Figure 7A), the level of GPX was very high, which is required for effective detoxification of ROS and maintaining their levels at a less harmful concentration [115]. This step results in the increased levels of the oxidized form of GSSG, but this process is dynamic in the glutathione cycle, and GSSG can be reduced back to GSH based on GR enzyme activity. The role of GR in the pool of other antioxidative enzymes in the GSH cycle and metabolism was also well demonstrated in chickpea plants under salinity [116] and drought [26], as well as in the developing embryos of *Arabidopsis* mutants [117]. With a lower level of GR in the chickpea accessions with the *CaABCC6* haplotype A, the GSH-reducing process is carried out more slowly, and it could increase the level of GSSG in cells. In the current study, this was exactly what happened under conditions of salinity, where the ratio of GSH/GSSG was significantly smaller in chickpea accessions with the *CaABCC6* haplotype A (Table 2). Additionally, the GST enzyme level was not so high in chickpea accessions with the *CaABCC6* haplotype A, and this is in consensus with lower expression of *CaABCC6* in the leaves of chickpea accessions with haplotype A under NaCl application (Figure 6). The non-effective relocation of glutathione conjugates from the cytosol to vacuole with a low activity of GST and the expression of *CaABCC6* in haplotype A completes the present overview.

Similar results were reported for tobacco plants (*Nicotiana tabacum* L.), where overexpressing two GST genes from common bean (*Phaseolus vulgaris* L) improved their tolerance to oxidative stress [118]. The authors provided compelling evidence that fresh weight and the number of leaves were significantly higher and associated with stronger expression. More *ABC* and antioxidative genes being activated in transgenic tobacco plants grown in heat and combined heat and drought conditions compared to WT.

In contrast, Figure 7B shows the proposed schematic diagram for chickpea accessions with the *CaABCC6* haplotype D2, and our claim deals with plants of chickpea cultivars Krasnokutsky-123 and Looch that are better adapted to oxidative stress. In the mature seeds of plants with haplotypes D2, the GPX protein level was significantly lower, producing the less oxidized form of GSSG, but there was much higher activity of GR, reducing GSSG to GSH very effectively. Additionally, glutathione conjugates can be relocated faster and stored in vacuoles due to the results that showed a high activity of GST in the seeds and a very high expression of *CaABCC6* in the leaves of haplotype D2 under salt stress (Figure 7B). Therefore, chickpea plants with *CaABCC6* haplotype D2 manage better with oxidative stress.

Finally, the presence of one mutation in the glutathione pathways can have a dramatic effect on plant growth and development, which can be demonstrated using the example of transgenic *tt19* mutant *Arabidopsis* plants with a defective *anthocyanin-related GST* gene [119]. The authors showed the results of a complementary test overexpressing the glutathione S-transferase gene, *GhGSTF12*, from cotton (*Gossypium hirsutum* L.), restoring a completely red leaf color in transgenic *Arabidopsis* seedlings with full recovery of glutathion conjugate translocation into vacuoles. A similar case with one single mutation in the SNP-1 position of the *CaABCC6* gene in the D2 haplotype of chickpea, as shown in the present study, may have a big impact on salinity tolerance, and we hypothesize that it may happen via better management of oxidative stress.

## 5. Conclusions

The present results support the hypothesis that chickpea plants with the *CaABCC6* D2 haplotype performed better in conditions of salinity stress. The measured parameters included low LN, high chlorophyll content, low Na^+^ accumulation and Na^+^/K^+^ ratio, low MDA content, and high GSH/GSSG ratio in the leaves of NaCl treated plants compared to other haplotypes. However, the gene *CaABCC6* is just one of four identified through marker–trait association analysis, and it is very likely related to better plant management of oxidative stress rather than sodium toxicity or osmotic components of salt application. The other three candidate genes identified in this study may also be important and have potentially critical impacts on plant photosynthesis and other plant traits related to salinity tolerance. Therefore, only the coordinated regulation of important genes can lead to an integrative plant response to salinity, and one example of chickpea plants with the *CaABCC6* D2 haplotype was shown here.

## Figures and Tables

**Figure 1 biomolecules-14-00823-f001:**
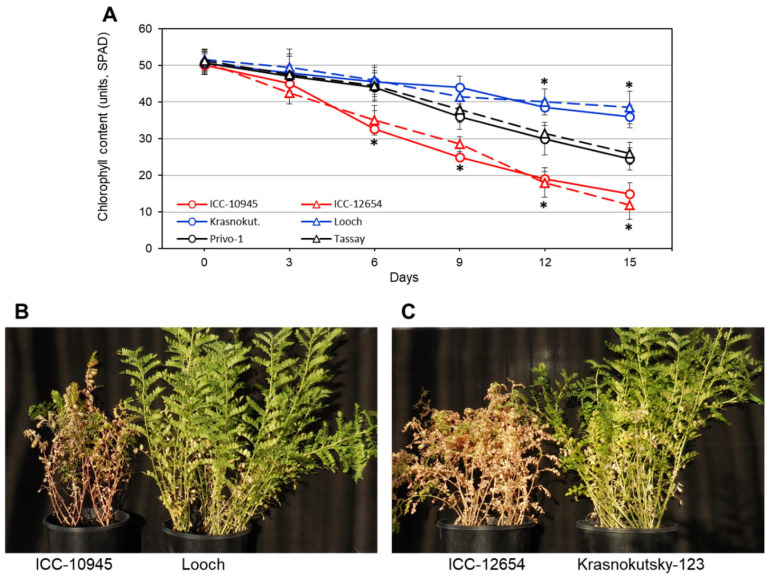
(**A**) Dynamics of chlorophyll degradation in plants of six selected chickpea accessions grown under 100 mM NaCl over 15 days. Significant differences are calculated with *t*-test and indicated by asterisks with at least *p* < 0.05. (**B**,**C**) Images of chickpea plants of accessions and cultivars with the most contrasting reactions to salinity, 100 mM NaCl, over 21 days.

**Figure 2 biomolecules-14-00823-f002:**
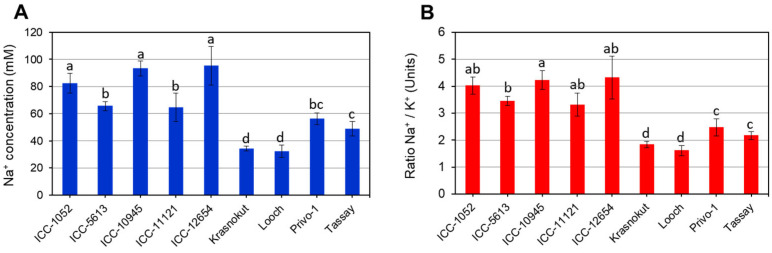
Accumulation of Na^+^ (**A**) and calculated Na^+^/K^+^ ratio (**B**) in leaves of chickpea plants after 100 mM NaCl treatment over 15 days. Significant differences were calculated with *t*-test and indicated by different letters (*p* < 0.05).

**Figure 3 biomolecules-14-00823-f003:**
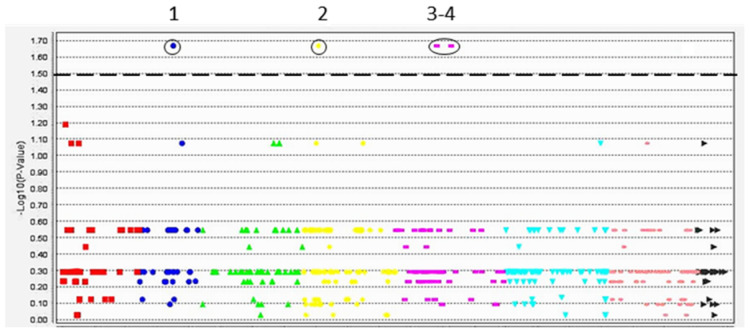
An example of Manhattan plot of significant *p*-value for marker–trait association (MTA) between 1600 studied DArT markers and leaf necrosis in chickpea accessions after plant treatment with 100 mM NaCl for 15 days, using the MLM model of TASSEL-5 program. Data are presented as dots with different colors and shapes in the order of chromosomes on the physical map of chickpea cv. Frontier, v.1. Four identified MTAs are circled and numbered on the top. Dashed line indicates the significant level of the associations (*p* < 0.05) for all four studied traits.

**Figure 4 biomolecules-14-00823-f004:**
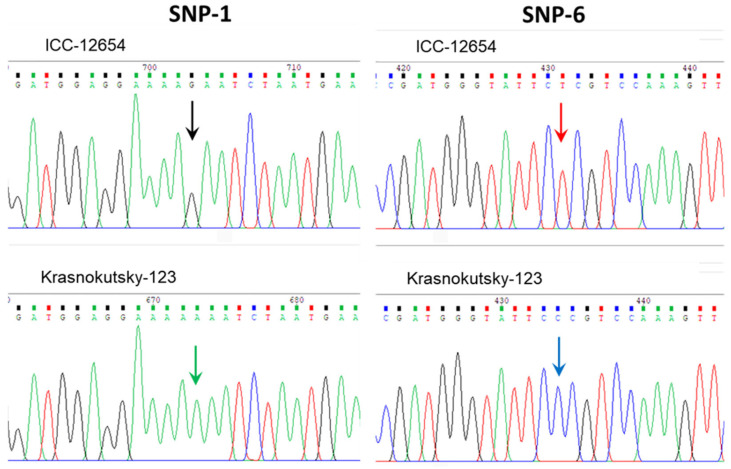
Fragments of Sanger sequences of SNP-1 and SNP-6 from seven SNPs identified in the *CaABCC6* gene on the example of chickpea accessions ICC-12654 and cv. Krasnokutsky-123. The positions of SNPs are indicated by colored arrows.

**Figure 5 biomolecules-14-00823-f005:**
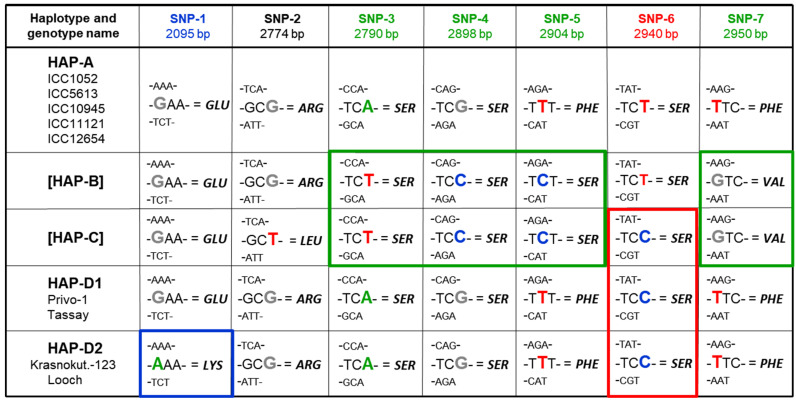
Identification of seven SNPs in the coding region of the *CaABCC6* gene (Ca09705), shown in different colors depending on haplotypes and accompanied by distance positions on the genome from the ‘start-codon’ of the transcript. Three earlier described haplotypes (A, B, and C) were retrieved from [77] for comparison. Two novel haplotypes (D1 and D2) are proposed in the current study. Boxes framed with various colors show differences between haplotypes. Triplets with SNP are shown in bigger font with encoded amino acid residues indicated with standard abbreviations to show synonymous or non-synonymous SNP. Full sequences representing the studied haplotypes are presented in the Appendix A.

**Figure 6 biomolecules-14-00823-f006:**
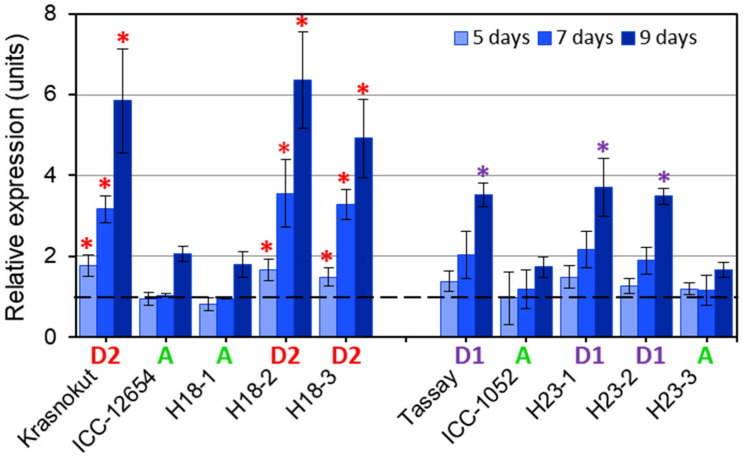
RT-qPCR expression analysis of the *CaABCC6* gene in parents and three selected F_6_ breeding lines from each of two chickpea hybrid populations, in plants grown in pots with soil after 15 days of 100 mM NaCl treatment. On the left-hand side, plants of hybrid 1 (♀Krasnokutsky-123 × ♂ICC-12654) and their breeding lines H18 are shown with designated *CaABCC6* haplotypes D2 (in red) and A (in green), respectively. Similarly, on the right-hand side, plants of hybrid 2 (♀Tassay × ♂ICC-1052) and their breeding lines H23 were designated as haplotypes D1 (in purple) and A (in green), respectively. Four consecutive time points were used for sampling, and controls were arranged as ‘point 0’ and set as unit level 1, indicated by the dashed line. Bars showing data for 5, 7, and 9 days after the start of salinity stress represent units of relative expression compared to controls. Expression data were normalized using the reference gene *CaELF1α* (elongation factor 1-alfa) and are shown as bars for average ± SE of three biological replicates (individual plants) and two technical repeats for each genotype and time point of NaCl treatment. Significant differences (* *p* < 0.05) between plants with different haplotypes for hybrids 1 and 2 were calculated using paired *t*-test and shown only for haplotypes D2 and D1 (but not for haplotype A) by red and purple asterisks, respectively, for each corresponding time point.

**Figure 7 biomolecules-14-00823-f007:**
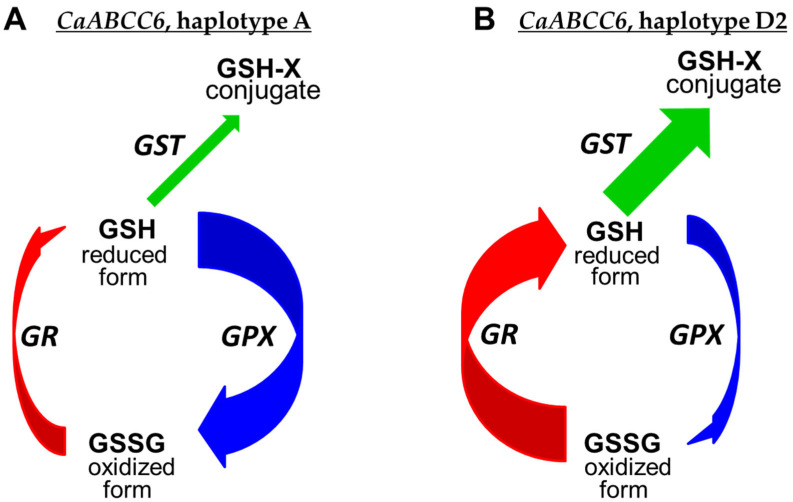
Schematic presentation of the *CaABCC6* gene involvement in the glutathione cycle and pathway in chickpea plants, in the examples of haplotypes A (**A**) and D2 (**B**). Abbreviations: Glutathione in reduced form (GSH), glutathione disulfide in oxidized form (GSSG), and glutathione conjugate (GSH-X) are indicated in regular font. Glutathione peroxidase (GPX), glutathione reductase (GR), and glutathione S-transferase (GST) are indicated in italics. The transitions between compounds catalyzed by these enzymes are indicated by color arrows. The width of the arrows corresponds to high or low activity of the enzyme, respectively.

**Table 1 biomolecules-14-00823-t001:** Leaf necrosis (LN) and chlorophyll content in leaves of one-month old plants of nine chickpea accessions exposed to 100 mM NaCl over 15 days. Average and SE are calculated based on three biological replicates (plants) in different pots. Significant differences among genotypes were calculated based on two-way ANOVA and indicated by different letters with at least *p* < 0.05.

Accession	Leaf Necrosis, Rate		Chlorophyll Content, Units-SPAD	
	Average	SE	Average	SE
ICC-1052	7.8 ^a^	0.3	13.5 ^a^	1.9
ICC-5613	7.6 ^a^	0.4	12.8 ^a^	2.0
ICC-10945	8.4 ^a^	0.5	15.2 ^a^	2.1
ICC-11121	7.8 ^a^	0.3	13.6 ^a^	2.2
ICC-12654	8.0 ^a^	0.4	12.3 ^a^	2.8
Krasnokutsky-123	4.6 ^c^	0.5	36.2 ^c^	2.6
Looch	5.0 ^c^	0.4	38.5 ^c^	3.1
Privo-1	6.2 ^b^	0.3	24.5 ^b^	2.2
Tassay	6.4 ^b^	0.4	26.2 ^b^	2.3

**Table 2 biomolecules-14-00823-t002:** Content of malondialdehyde (MDA) and ratio between reduced glutathione (GSH) and oxidized glutathione (GSSG) in leaves of one-month old plants of nine chickpea accessions after 100 mM NaCl treatment for 15 days. Average and SE are calculated based on three biological replicates (individual plants in different pots). Significant differences among genotypes were calculated based on two-way ANOVA and indicated by different letters with at least *p* < 0.05.

Accession	MDA, nmol/mL		GSH/GSSG Ratio	
	Average	SE	Average	SE
ICC-1052	99.2 ^a^	7.2	8.6 ^a^	0.7
ICC-5613	87.9 ^a^	5.5	7.9 ^a^	1.1
ICC-10945	78.3 ^a^	6.1	7.2 ^a^	1.3
ICC-11121	89.2 ^a^	6.9	8.7 ^a^	0.8
ICC-12654	73.6 ^a^	6.4	6.9 ^a^	1.2
Krasnokutsky-123	35.5 ^c^	2.4	16.2 ^c^	2.1
Looch	32.3 ^c^	2.6	15.5 ^c^	1.8
Privo-1	56.4 ^b^	4.2	12.1 ^b^	1.2
Tassay	54.9 ^b^	4.1	11.4 ^b^	1.0

**Table 3 biomolecules-14-00823-t003:** The significant marker–trait associations (MTA) identified using 1600 effective markers after filtration of 6K DArT assay analysis for chickpea accessions with all four studied traits related to salinity and oxidative stress responses with MLM models using the TASSEL-5 program. Leaf necrosis (LN) and Na^+^ accumulation (Na^+^) traits were related to salinity stress, whereas the content of malondialdehyde (MDA) and GSH/GSSG ratio (GSH) were indicators for oxidative stress. The identified DArT markers for MTA and their positions in the reference genome of chickpea, cv. Frontier, v.1, were used for the identification of the closest candidate genes with their annotations and positions of the same reference genome. The significance level of the associations is indicated as follows: * 0.01 < *p* < 0.05; and ** 0.001 < *p* < 0.01. The gene *Ca09705*, indicated in Bold, was selected for further analyses.

Chromosome	DArT Marker and Position on Reference Genome	*p*-Value for MTA, Four Traits	Closest Candidate Gene and Position on Reference Genome	Annotation of Candidate Gene
**Ca2**	**10265114** 35,466,143–35,466,206	LN = 0.0076 ** Na^+^ = 0.025 * MDA = 0.0065 ** GSH = 0.0029 **	**Ca09705** 35,445,660–35,452,426	**ATP-binding cassette,** **ABC transporter**. GO:0005524 (ATP binding)
Ca4	10259848 10,526,726–10,526,760	LN = 0.0062 ** Na^+^ = 0.046 * MDA = 0.018 * GSH = 0.0082 **	Ca04289 10,526,277–10,528,094	Pentatricopeptide repeat (PPR) superfamily protein. GO:0005515 (protein binding)
Ca5	23885575 23,699,770–23,699,833	LN = 0.0065 ** Na^+^ = 0.028 * MDA = 0.0076 ** GSH = 0.012 *	Ca17680 23,697,998–23,700,690	Heat shock protein 81.4. GO:0005524 (ATP binding)
Ca5	10265729 43,890,086–43,890,149	LN = 0.0025 ** Na^+^ = 0.042 * MDA = 0.0085 ** GSH = 0.017 *	Ca12664 43,878,837–43,880,175	Photosystem II 22 kDa protein. IPR022796 (Chlorophyll A-B binding protein)

**Table 4 biomolecules-14-00823-t004:** Mass spectrometry analysis of glutathione peroxidase 3 (GPX-3), glutathione reductase, chloroplastic/mitochondrial (GR-chl/mit), and glutathione S-transferase L3 (GST-L3) accumulation in mature seeds of studied chickpea accessions using spectral library for fractions and unfractionated samples. Each sample (Group 1) was compared with a single selected genotype of chickpea cv. Looch (Group 2) and presented in both percentage of change and Log2 ratio. Three biological replicates (plants) and two technical runs using Fusion Lumos/Exploris 480 mass spectrometry were applied for the protein analysis, identified in chickpea UniProt database. Significant differences are indicated as the corrected *q*-values as follows: * 0.01 < *q* < 0.05; and ** 0.001 < *q* < 0.01.

Comparison (Group 1/Group 2)	GPX-3 (Q8L5Q6)		GR-chl/mit (A0A1S3E1Y6)		GST-L3 (A0A1S2XSM5)	
	% Change	Log2 Ratio	% Change	Log2 Ratio	% Change	Log2 Ratio
ICC-1052/Looch	184.4 **	1.5	−85.2 **	−2.8	−42.4 *	−0.8
ICC-5613/Looch	179.5 **	1.5	−48.2 *	−0.9	−37.1 *	−0.7
ICC-10945/Looch	288.2 **	2.0	−64.0 **	−1.5	−55.7 *	−1.2
ICC-11121/Looch	236.9 **	1.8	−91.6 **	−3.6	−73.6 **	−1.9
ICC-12654/Looch	272.9 **	1.9	−75.7 **	−2.0	−38.1 *	−0.7
Privo-1/Looch	87.8 *	0.9	−41.9 *	−0.8	−50.9 *	−1.0
Tassay/Looch	108.6 *	1.1	−47.4 *	−0.9	−47.9 *	−1.0

## Data Availability

The data presented in the manuscript are available in the Appendix A.

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
