# Peer review of "Haplotypes of ATP-Binding Cassette CaABCC6 in Chickpea from Kazakhstan Are Associated with Salinity Tolerance and Leaf Necrosis via Oxidative Stress"

_biomolecules, 2024, doi:10.3390/biom14070823_

Round 1

Reviewer 1 Report

Comments and Suggestions for Authors

The authors of this manuscript evaluated salinity tolerance in chickpea accessions from a germplasm collection and in cultivars from Kazakhstan, and divide into different tolerant groups based on the tolerance performance. They find that the chlorophyll content, leaf necrosis (LN), Na+ accumulation, malondialdehyde (MDA) content and glutathione ratio GSH/GSSG was significantly associated to the salinity tolerance of the evaluated chickpea accessions. They further revealed the presence of four possible candidate genes in the chickpea genome which may be associated with the three groups by Marker-trait association (MTA) between 6K DArT markers and four traits (LN, Na+, MDA and GSH/GSSG) and finally find that the haplotypes of ATP binding cassette, CaABCC6, in Chickpea from Kazakhstan are associated with salinity tolerance and leaf necrosis via oxidative stress.

The experiments were overall properly performed and the methods, data and analysis adequately test the hypothesis and the results and conclusions are reasonable. It has scientific value in study salinity tolerance and thereafter molecular breeding in chickpea. It can be considered for publication after some revisions or explanations listed below:

1. The number of chickpea accessions for the research is seemly somewhat not so enough in the manuscript, please give credible explanation for the choosing of materials! It suggests to confirm the key SNP and the novel haplotype of the CaABCC6 using more cultivars and germplasm of chickpea!

2. The “Introduction” part is somewhat long-winded, it suggests to slightly simplify this part!

3. Some details presented in the text and figures and tables should be carefully checked to ensure accuracy! e.g., in Abstract, line 25, “malondialdehyde” should be “malondialdehyde content”!

Author Response

  1. The number of chickpea accessions for the research is seemly somewhat not so enough in the manuscript, please give credible explanation for the choosing of materials! It suggests to confirm the key SNP and the novel haplotype of the CaABCC6 using more cultivars and germplasm of chickpea!

Response: The choice of the studied chickpea accessions was based on very diverse reactions of plants in the preliminary experiments during the hybridization program. We inserted this statement in the M&M section (L175-177). At the same time, we agree with the Reviewer and we are planning to extend this study using more chickpea cultivars and accessions. We also added a sentence about this statement in the Discussion section (L675-677).

  1. The “Introduction” part is somewhat long-winded, it suggests to slightly simplify this part!

Response: We have slightly shortened the Introduction section (L50-51, L103, L105, L107, L112-115, L123-124) but it was not easy because our manuscript covered very different areas of study, from salinity to oxidative stress, DArT, marker-trait association, candidate gene, SNP and haplotype genotyping, gene expression and proteomics. All these parts must be described properly and logically in the Introduction section. Therefore, this is very hard to reduce the Introduction section without loss of meaning and order of the presentation.

  1. Some details presented in the text and figures and tables should be carefully checked to ensure accuracy! e.g., in Abstract, line 25, “malondialdehyde” should be “malondialdehyde content”!

Response: The manuscript (including text, Figures and Tables) has been checked. Corrections were made and indicated in the revised manuscript for accuracy and grammatical improvement.

Reviewer 2 Report

Comments and Suggestions for Authors

The study covers an interesting area of research. Comparing chickpea varieties for their tolerance to salt stress is important and therefore the study has merit. I recommend revision before publication. Below are my comments

1.     In the introduction, the authors have introduced various techniques used to identify differences in chickpea varieties tolerant to salt and drought stress. Since this paper is not a review article, summarizing different techniques is beyond its scope. The introduction should instead focus on the salt stress problem in Afghanistan, its impact on agricultural productivity, the effects of high salt concentrations on plant physiology and productivity, the number of chickpea cultivars that can be grown in the region, and the salt tolerance capacity of these cultivars.

2.     In line 72, the manuscript mentions a 60mM NaCl treatment, but later sections refer to a 100mM NaCl treatment. To ensure clarity and consistency, please confirm the correct NaCl concentration used in the study and standardize this throughout the manuscript.

3.     The term "mass spectroscopy" used in line 137 is incorrect. The correct term is "mass spectrometry."

4.     In line 172, the phrase "and distributed for research purposes to Australia" is unclear. Please explain.

5.     In the manuscript, SPAD meter is used to measure chlorophyll content. The SPAD meter can show variations between chickpea varieties due to differences in leaf structure, thickness, and chlorophyll distribution. Did the authors perform a species-specific calibration of the SPAD meter to account for these potential variations?

6.     What is the purpose of MDA content measurement with and without TBA ?

7.     In the Materials and Methods section, authors do not provide details of how DArTseq data was processed. Adding details of data processing steps will provide clarity to the readers and add value to the paper.

8.     In order to compare chickpea varieties in their response to salt and oxidative stress, authors have compared seed protein and talk about protein related to glutathione pathway. Why choose analyzing protein content in seed instead of glutathione in plants subjected to salt stress?

9.     Identifying chickpea varieties relatively tolerant to salt stress and oxidative damage is important for improving crop productivity under stressed conditions. However, an essential aspect to consider is also the productivity under stress, did authors also look into seed setting and seed quality in the chickpea varieties when growing in 100mM NaCl ? Comparing this will add to the quality of the paper.

10.  Could the authors clarify why only one gene, Ca09705 (CaABCC6), out of the four identified MTAs was chosen for further analysis? Is this decision based solely on its association with glutathione transport?

Comments on the Quality of English Language

English language needs improvement 

Author Response

  1. In the introduction, the authors have introduced various techniques used to identify differences in chickpea varieties tolerant to salt and drought stress. Since this paper is not a review article, summarizing different techniques is beyond its scope. The introduction should instead focus on the salt stress problem in Afghanistan, its impact on agricultural productivity, the effects of high salt concentrations on plant physiology and productivity, the number of chickpea cultivars that can be grown in the region, and the salt tolerance capacity of these cultivars.

Response: We agree with the Reviewer and the Introduction section was slightly shortened (L50-51, L103, L105, L107, L112-115, L123-124). However, it was not easy because our manuscript covered very different areas of study. Therefore, a more extensive reduction of the Introduction section was not possible without loss of meaning and order in the presentation.

  1. In line 72, the manuscript mentions a 60mM NaCl treatment, but later sections refer to a 100mM NaCl treatment. To ensure clarity and consistency, please confirm the correct NaCl concentration used in the study and standardize this throughout the manuscript.

Response: There is no conflict between the statements about 60 and 100 mM NaCl. The experiment with 60 mM NaCl treatment of 557 chickpea accessions in hydroponics subsequently assessed for leaf necrosis was described in the Introduction section (L71) with the corresponding reference ‘[13]’. In our experiments, described in M&M (L193) and Results sections (L347, L358, L363-364 and other), 100 mM NaCl was applied to chickpea plants grown in soil. There is no relationship between experiments published in ‘[13]’ and those presented in our study. Therefore, we cannot make any correction to the different levels of NaCl applications in non-related experiments.

  1. The term "mass spectroscopy" used in line 137 is incorrect. The correct term is "mass spectrometry."

Response: The term was corrected (L129).

  1. In line 172, the phrase "and distributed for research purposes to Australia" is unclear. Please explain.

Response: This phrase was deleted to simplify a clear meaning (L164).

  1. In the manuscript, SPAD meter is used to measure chlorophyll content. The SPAD meter can show variations between chickpea varieties due to differences in leaf structure, thickness, and chlorophyll distribution. Did the authors perform a species-specific calibration of the SPAD meter to account for these potential variations?

Response: Yes, we agree with the Reviewer, and we made preliminary measurements by SPAD-meter to estimate chlorophyll content in leaves of all studied accessions in different leaves and stages of plant development. A sentence was added in the M&M section addressing this issue (L205-207).

  1. What is the purpose of MDA content measurement with and without TBA ?

Response: In this method, MDA content was measured with and without TBA, a requirement designed to increase the accuracy of MDA measurements by correcting for any non-MDA compounds which also absorb at 532 nm, such as anthocyanins. This sentence was inserted in M&M section (L235-237).

  1. In the Materials and Methods section, authors do not provide details of how DArTseq data was processed. Adding details of data processing steps will provide clarity to the readers and add value to the paper.

Response: One sentence was inserted in M&M (L265-267) about filtrations of SNP and DArT markers to address this point.

  1. In order to compare chickpea varieties in their response to salt and oxidative stress, authors have compared seed protein and talk about protein related to glutathione pathway. Why choose analyzing protein content in seed instead of glutathione in plants subjected to salt stress?

Response: The Reviewer is absolutely right and we completely agree with this statement. Our explanation is very simple. Initially, there were two independent experiments from different projects: (1) proteomic analysis in chickpea seeds and (2) study of plants under salinity stress; and we did not plan to combine it together. However, after analysis of the results, we found that a fragment with proteomic profiling results might be interesting and suitable to illustrate our hypothesis about enzymes controlling glutathione cycles in the studied chickpea genotypes. Therefore, we decided to add the proteomics results in chickpea seeds in the Discussion for better and logical explanation of a potential role of the CaABCC6 gene in glutathione conjugate in the response to oxidative stress. A fragment addressing this issue was inserted in L699-704.

  1. Identifying chickpea varieties relatively tolerant to salt stress and oxidative damage is important for improving crop productivity under stressed conditions. However, an essential aspect to consider is also the productivity under stress, did authors also look into seed setting and seed quality in the chickpea varieties when growing in 100mM NaCl ? Comparing this will add to the quality of the paper.

Response: We observed the difference in seed setting and yield at harvest in the studied chickpea accessions after NaCl application. However, three plants (biological replicates) were not sufficient to make a conclusive report on seed yield, therefore this must be assessed in a separate study in future with a greater number of plants. A fragment was inserted in Discussion section (L671-675) addressing this point.

  1. Could the authors clarify why only one gene, Ca09705 (CaABCC6), out of the four identified MTAs was chosen for further analysis? Is this decision based solely on its association with glutathione transport?

Response: As we wrote in the Results section in our manuscript (L448-450), “Only one gene Ca09705 (CaABCC6) was selected for further analyses based on published information that it was involved in glutathione transport in chickpea plants in non-stressed conditions [77]”. It was impossible to study all four candidate genes because it could overload the manuscript and, therefore, only one gene was selected. However, in the Discussion section (L615-617), we wrote: “It is very likely that all identified candidate genes are important, acting together for better plant response and adaptation to stressful conditions”. In this regard, we added that ‘all of them are planned for our further study’ (L617-618).
